# Keeping Up with Caputo: Of Specters and Spooks—Transcendence and Happiness in Caputo's Radical Theology

**Joeri Schrijvers**

School of Philosophy, North-West University Potchefstroom, Potchefstroom 2520, South Africa;
schrijversjoeri@gmail.com

**Abstract:** This essay serves two purposes. First, it wants to introduce readers to John D. Caputo's Radical Theology by way of his recent *Specters of God* (2022) in which his radical theology truly comes to fruition. The essay provides in this introduction through elucidating this recent work and by pointing to earlier discussions of similar themes and figures in Caputo's corpus. It will be shown that Caputo's work is a genuine contemporary search for transcendence, asking all the right questions at the right time. Recently, for instance, Caputo is asking what becomes of the human search for meaning if this entire cosmos is destined to fade away in a Big Crunch. Second, this essay wants to critically address some remaining unclarities in this radical theology. It is to be noted, for instance, that at crucial stages Caputo repeats some aspects of the thought of divinity in theism that he nonetheless says he wants to overcome; at these stages, then, 'God' is allowed to be an exception to the worries of the world after all. This essay, too, wants to investigate Caputo's rather enigmatic insistence of the possibility of joy and happiness in a mortal, finite world that would celebrate only a finite, mortal God. That finitude, instead of lasting and eternal salvation, serves as the very condition of possibility of true joy is an unexamined axiom running through Caputo's recent works. In this regard, this essay wants to point to both the beauty but also the frailty of this thought.

**Keywords:** John D. Caputo; radical theology; metaphysics; transcendence; joy; deconstruction; happiness





## 1. Introduction

It is *kind of* hard to keep up with Caputo. The publication of his recent *Specters of God. An Anatomy of the Apophatic Imagination* (2022) will be a surprise to not a few of Caputo's readers. The book's length is considerable; in about 400 pages, Caputo discusses authors that have not figured much in his quite considerable corpus, such as Schelling or Tillich. Caputo wants to show the historical background of his project in radical theology—where this radical theology is coming from and where it, today, might be going.

Yet one might have expected that, after the somewhat reluctant start of this radical theology, with his "coming out of the closet as a theologian" in *The Weakness of God. A Theology of the Event* of 2006, the follow-up with *The Insistence of God. A Theology of Perhaps* of 2013, and the completion of the trilogy in 2015 with the publication of *The Folly of God. A Theology of the Unconditional*, Caputo might perhaps have said the final word about his projects in theology. To this constatation it could be added that, aside from this theological project, Caputo has made several intriguing, more philosophical side-steps with the publication of a book on truth (in 2013) and an introductory but no less intriguing volume on *Hermeneutics. Facts and Interpretation in The Age of Information* in 2018. The bite of theology did not disappear though; in 2019 *Cross and Cosmos. A Theology of Difficult Glory* followed the trilogy mentioned above, itself following, aimed at a more general audience, *Hoping against Hope. Confessions of A Postmodern Pilgrim* in 2015. Recently, some essays previously scattered around in scientific journals were gathered in his *In Search of Radical theology* (2020).

What more could there be to say? Yet what should be noted from the outset is that the work, Caputo's work, is never done. On the contrary, we find Caputo beginning over time and again, relaunching his theses, communicating and elucidating what he wants to say time and time again, and so commencing anew. We should note, then, how *hospitable* this thought is—it does not stop dialoguing with its audience (to which my own contribution to his radical theology might testify—here Caputo responds to each single essay in a volume gathering European scholars around his thought (Schrijvers and Koci 2022)—but also, and especially, the fact that Caputo, prolific a writer as he is, does not stop dialoguing with "everyone who is anyone" in the contemporary philosophical field.

Hospitality is obviously a word of Derrida. Yet it can safely be said that Heidegger is more present in the book that is at the core of this essay. Caputo here makes his way all the way from Aquinas and Eckhart—the writers with which Caputo's work, back in 1982 with the publication of *Heidegger and Aquinas. An Essay on Overcoming of Metaphysics* and in 1978 with *The Mystical Element in Heidegger's Thought*, pretty much started—up to an interpretation of contemporary physics' relevance for theology, through a detour via (mainly) Schelling and Tillich. In a sense, Caputo here points to that which blocked radical theology's moving to the forefront of contemporary theological thinking. Caputo here offers what one can call a (not so) concise history of radical theology by spelling out what in his earlier work remained somewhat implicit, such as the shift from Hegel's *Vorstellung* and Tillich's ontology to hauntology and spectral theology (Caputo 2020, p. 22).

Theology's bite perhaps comes down to one simple question, and it is this question, it seems, that keeps Caputo going even after his *The Insistence of God*: "Will it preach?". This question taps into a concern that Caputo very early on mentioned already in what he calls the "prolegomena" (Caputo 2020, p. 3) to his radical theology. It is in this book, the books that sets out the contours of a "religion without religion", that he speaks of a "generalized apophatics" (Caputo 1997, p. 28) which, now, can nicely be linked to the "apophatic imagination" on which his most recent book turns. It should also be noted that Caputo regularly insists on the continuity of his works; radical hermeneutics in a sense already was a radical theology (Caputo 2018a, p. 19).

In 1997, however, Caputo's general concern read this way:

> "I will argue for what I will call here a 'generalized apophatics,' by which I mean that negative theology [at the time a heated debate on Derrida's deconstruction and negative theology was happening, Caputo's take is that there is an element of non-knowing in religion that needs to be teased out, JS] is everybody's business, that it has a general translatability [ . . . ] A deeply affirmative desire for something that is always essentially other than the prevailing regime of presence [ . . . ] is of general interest. A passion for the impossible is of general concern". (Caputo 1997, p. 28)

Since then, the intuition that his work on radical theology matters for people outside academia never left Caputo (e.g., Caputo 2013, p. 104; 2020, p. 24). And what *Hoping against Hope is* for his radical theology, the book *On Religion* is for his work on Derrida's religion without religion: an outbreak out of the academic confines, explaining the general ideas to the broader public. We argue that it is onto this concern that the question "will it preach?" grafts itself a question that obviously haunts Caputo and unsettles his works; we will see him coming back to it again and again, even in his most recent work. This question whether this radical theology is suitable at all for the people in and of the church is, to my knowledge, first posed (and published) in 2015 by Brian McLaren (McLaren 2015, pp. 118–223) and then repeated by Peter Rollins in his Foreword to *Hoping against Hope* (Caputo 2015a, p. ix). It is quite likely that this question, which equally has never left Caputo (see e.g., Caputo 2015a, p. 115; 2020, p. 73), sparked writing *Hoping against Hope*.

## 2. Of Specters and Spooks

"I am saying that God does not exist. *Tout court*" (Caputo 2022a, p. 41). One can hardly imagine a more radical sentence from a theologian's pen. The question whether this

radical theology can be welcomed in the church and in religious communities thus rightly deserves to be posed: how on earth can one preach that God does not exist? It is, in this sense, understandable that Caputo's latest takes the time to rehearse the principles and premises of his radical theology one more time.

*2.1. The Principles and Premises of Radical Theology*

The books set off with an elaborate take on an interpretation of the phenomenological reduction when it comes to radicalizing theology. The label "radical theology" tells us at least two things already, first, that it is indeed a theology—although *Specters of God* can be interpreted as the more philosophical background of the earlier theological trilogy— and, second, that it is "radical". It is radical and theological in a special sense, however. By "radical", Caputo does not mean that it goes to the "roots" or the most fundamental "foundation" of theology. Rather, it seeks what rages through classic theologies and confessional traditions in order to open these up to the event that made them possible in the first place. For this, "radical theology" is first of all a *radicalization* of precisely these traditions. Radical theology so exposes the confessional traditions to what is going on "in" them, but what is not "of" them—religion, *Specters of God* tells us, is claimed *by* it knows not what but cannot claim *to* own the event that first made it happen (Caputo 2022b, p. 229).

This is exactly what makes radical theology a theology: theology, too, is disturbed from within by God knows what—which Caputo has earlier labeled "event" and a "call" and, in this book, as "feeling about [its] apophatic core (Caputo 2022b, p. xii)—but the very fact that there is such disturbance and such groundlessness "is itself theological" (Caputo 2020, p. 4). It, in effect, testifies to an infinity that no finite structure or culture can contain. And it is for such "infinitival exaction" (ibid., p. 13) that radical theology pleads, for while it *presences*, perhaps, in each confessional religion, it is not to be confused with the *essence* of this religion—an error that has confused many of Caputo's commentators. Radical theology is an attempt to radicalize these religions in order to come into contact with what moves them, what keeps them going, and sparked them from the very beginning. Yet this theology remains radical in a more usual sense too: it concerns a theology, different from the confessional theologies, it is "a theology worthy of the name" (Caputo 2022b, p. xiii)—a theology about that which incites our theologizing in the first place—and "a theology that thinks" (ibid.), one, that is, awoken "from its dogmatic-theistic slumbers" (Caputo 2022b, p. 203). With this, we believe, Caputo puts himself at the vanguard of contemporary theology.

At times, Caputo sounds like a postmodern Hans Blumenberg; where the latter says that modernity offered a new problematic to thought to which the old theological answers could no longer respond, Caputo states that "the radical in radical thinking implies [ . . . ] that we raise a theological question with no guarantee that we will get a theological answer [ . . . ] and, on the other hand, that we raise a philosophical question with no guarantee that we will not get a theological answer" (Caputo 2022b, p. 41). Caputo's quest into the apophatic imagination, beyond Aquinas, beyond Eckhart, beyond Schelling, beyond Tillich, will put as at the brink of just that: "a nontheological reply to a theological inquiry" (Caputo 2022b, p. 72). Except that Caputo will go a long way with the thinkers of our day to come up with at least the beginning of a theological reply.

Caputo's radical theology thus attempts to return theology to its "proper element" and its "proper truth" (Caputo 2022b, p. 17, 21). We need to learn, in short, to theologize about the ordinary world, for "the apophatic is the stuff of everyday life, found on every register, in every place" (Caputo 2022b, p. 7). This, in effect, is the stated aim of Caputo's latest book, namely, "to prompt a spirited search for truth" (Caputo 2022b, p. 5) in order that the spirited search *of us all* comes into view. It is this existential dimension of Caputo's work that rarely has been highlighted and that the current essay explicitly wants to highlight.

For this, however, for this quest, this search and this spirit, a reduction is needed. The reduction will take us from our beliefs to a more elemental faith. It attempts, as Caputo has it elsewhere, to "inhabit the distance" (Caputo 2018b, pp. 229, 294) between confessional

theology and radical theology, between what is conditioned—like religion—and what comes to us unconditionally. Radical theology is a radicalization of a particular religion just like deconstruction is always a deconstruction of something (cf. Caputo 2020, p. 3; 2022b, p. 244). The aim of such a deconstruction of Christianity is "its reinvention, or re-actualization, in the world" (Caputo 2018b, p. 180) and to so "open or re-open religion to its future" (Caputo 2020, p. 52), to see whether this thing called religion still has some time of life, for "to deconstruct something is not to destroy it, but to expose it to its future, which can turn out for the better unless it is for the worse" (Caputo 2022b, p. 271).

If we now, for Caputo, need to "live in this distance" (Caputo 2015a, p. 37), then perhaps the earlier work of the theological trilogy can be seen to forge this distance precisely, to open the gap between confessional religion and radical theology—hence why not a few commentators questioned this distance and this gap precisely. In his recent work, however, the distance is forged and Caputo calls on the reduction to show us how we can traverse the distance between confessional and radical theology and, perhaps, how we can *dwell* in this distance too: "everything turns on how to establish the distance, on how to open the space between [radical and confessional theology], which is how the radicalization is achieved. [Thus] far I have been describing this as a process of "weakening" which I will elaborate [ . . . ] as a "reduction" of confessional theology to radical theology" (Caputo 2020, p. 15).

Even though this reduction is no destruction, still some things will have to go. Yet this reduction, for Caputo, is not a phenomenological striptease to the "naked that it is" of religion: it does "not diminish" or "take away anything from us other than our illusions" (Caputo 2020, p. 20). It reduces, therefore, to a sort of remainder of and within religion, and we will later have to ask what exactly the status of such a remainder is: it concerns "the reduction of *x* but not without gaining access to a more radical *x*" (Caputo 2020, p. 20) as when Caputo finds through a reading of Mt. 25 "inside religion [ . . . ] a religionless Christianity" (Caputo 2015a, p. 56).

A religion, very much like a cultural narrative, Caputo will argue with Derrida, "may have an identity, [it] is not identical with itself. [It] is a complicated complex of different forces" (Caputo 2022b, p. 246). It is only when a religion or culture tries to identify with and as itself, to "remain the same always and everywhere" as a certain tradition in continental philosophy would have it, that the call of the other is silenced. The mark of the other—the fact that there is some other, others, and otherness in general—is so suppressed. From the very beginning, then, religion, as much as any other culture, would be stamped with otherness. The fact of being opened to something other than itself is a promise and a threat at the same time, a sentence to *life death* Derrida would say there is a chance for this culture to live on as much as the possibility exists that it will not live much longer. Whereas Caputo's *In Search of Radical Theology* spoke of a disturbance within these particular religions and cultures, a disturbance that moreover is actively "created" by radical theology as it is passively discovered in the name of God especially—since here there is "a concentration and an intensification of insinuating disturbances" (resp. Caputo 2020, pp. 3, 15)—for such a disturbance Caputo now uses the term autodeconstruction, present for instance in Eckhart's idea of a *Gottheit* beyond God. Such an idea, in effect, is "an autodeconstructive disturbance occurring *within* theology. [There] is a deconstructive force disturbing this edifying construction [ . . . ] from within" (Caputo 2022b, p. 66) to go on telling us, that

> "as a construction, a culture is inherently autodeconstructible, destabilized from within. But on the other hand, a culture is a tradition which means it gives or hands over [ . . . ] a legacy, a complex of promises and aspirations—in paradigmatic words like *justice, freedom,* and *democracy*—that make the present order restless with desire for what is coming. So the present is made restless by the past that promises a future [which] calls upon *us* to make this promise come true. That is where we do have a role to play". (Caputo 2022b, pp. 249–50)

Things are no less autodeconstructive in religion. Here, too, one will find a passive "deconstructibility from within", one "which is not our doing", and an active "a decon-



structibility from without" in and through which we make more of justice that has been the case until now, for instance. These two elements, then, will also be present in the radicalization of religion. One may, moreover, legitimately wonder whether the human condition itself is no autodeconstructive condition; there both is a deconstructibility from within, as we are all finite and mortal, and a deconstructibility from without, as the care for our being might always lead to change and transformation even in face of death.

Before, however, turning to these aspects of a more existential nature in Caputo, we will now briefly highlight the main stages of the reduction in Caputo's *Specters of God*.

### 2.2. An Inflection of Phenomenology: The Theopoetic Reduction

We will not delve too much into lengthy discussion of the book's main figures but read these chapters with an eye on just what needs to be radicalized in their respective thoughts according to Caputo. Careful readers of Caputo's *Specters of God* will quickly notice that there is somewhat of a phenomenological turn at issue. The reduction that is here elaborated comes with its own epoche and asks us right away to rid us of all beliefs in a supernatural entity that does things (or does not). Only then, Caputo states, "we shift into the theopoetic attitude, [suspend] the supernatural attitude and [bracket] the supernatural signified" in order to learn to see that revelation is not a message coming from on high but is rather to be seen as "the inbreaking of a new vision of life" (Caputo 2022b, pp. 8–9).

Quite quickly too, Caputo's existential emphasis comes to the fore. To let the "theopoetic phenomenon" show itself from itself, we must learn to let theology's founding texts "speak for themselves and from themselves" (ibid., pp. 18–19). This is not to say, naively, that we must read these texts with no background at all, or come to a unique interpretation of them; it is rather to say that we must allow ourselves to be gripped by these stories again, clothed and covered as they are by years and years of "religious" and "authoritative" interpretations. We, therefore, need let their poetics "exert a powerless power" over us, a power "that cuts through prosaic life and leaves us shaken, disturbed, and solicited, having revealed to us an alternate way to live" (ibid., p. 19), or at least that other ways to live are possible. Stories like these, Caputo concludes, "are not logically verified, but existentially witnessed to; they are not logically falsified either [but] can die off [ . . . ] if and when they lose their grip on us" (ibid.). This is not to say that we need to approach these texts ignorant of their contexts and established meanings; it is rather to recognize first that these texts *speak* to our lives before they will become or have become objects of study. If, for instance, these texts speak of the Kingdom of God, this Kingdom is not, for Caputo, a quasi-geographical description of a place somewhere up there nor the narration of some sort of chronology that will happen once this eon reaches its end: it is *kairological* entirely, speaking at the right time and the right place to show something to someone (ibid., p. 21). They "reveal another world" (ibid., p. 20) in the very precise sense that something else than the status quo is, was, and will be possible even when it seemed impossible.

Again, this reduction is not a critique of theology, it is to find out what has prompted us to theologize in the first place and what ended up in images that "[give] creative form or figure to something *not* of our own making" (ibid., p. 22), it is to poeticize and theologize about our being-in-the-world that was brought about without our proper accord (and of which we try to make sense with all the signifiers, high and low, available to us).

Such an *epoche* opens a discursive space, for Caputo, where believers and nonbelievers can meet: "theopoetics does not ask nonbelievers to believe in angels [ . . . ] but rather to suspend their disbelief (naturalism) [and it asks] believers to suspend their belief (supernaturalism)" in order to put "the apophatic imagination" in play and "the figures and the narratives of revelation are protected from being believed or disbelieved" (ibid., p. 17). The belief, based on transcendental reason, that science and technology are able to systematize and categorize the world in its entirety as well as the belief in the existence of highest being up there intervening (or not) down here must be bracketed. The *epoche* starts with an admittance of non-knowing, by admitting that none of our claims to truth are comprehensive.

The next step of this phenomenological inflection is "the reduction of a strong theology of the Supreme Being to a weak theology of the event" (ibid., p. 26), to seek for, as Caputo often has it, what might be "going on" in the name of God and in religion—what it is about and what prompted this search. Caputo proposes first an "ontological reduction", which he finds in the work of Paul Tillich. If we can no longer speak of God as a highest being, we might perhaps speak of being itself, about what it is to be and look for, as Tillich did, the ground of being to see what (and what not) is divine about being itself. One will see Caputo struggle with such a panentheism throughout the entire book, but he is clear that such an ontological reduction needs to be accompanied with a "second step" (ibid., p. 27), a hauntological reduction, of which Derrida offers the example. If the ontological reduction still has too firm a grip on being and beings—being able *knowingly* to state what a being exactly is for instance—it needs to be complemented once more with an element of apophaticism, with a "grammatological reduction" (ibid., p. 35) where our language, our narratives, our words are immersed in an endless play of naming these beings—and our being-in-the-world—with no chance of an ultimate and final name for Being/beings. Caputo is clear that such an atheism toward both God as the highest being and an agnosticism, ultimately, about Being's divinity, is but to seek the "beginning" of theology (ibid., p. 32). A radical theology might begin after the death of God as an independent entity and as the foundation or ground of any and all being/s.

### 2.3. Radicalizing Premodern Theology and Modern Philosophy

Caputo will then continue with a lengthy journey through the history and philosophy to see how this apophatic imagination plays in the history of thought. Caputo begins where his own journey began: with Eckhart and Aquinas. From Aquinas, Caputo takes the participation of beings in God: we all have received our esse from "the self-subsisting act of being" (ibid., p. 46) which, itself, is not limited by any particular form whatsoever. Being is everywhere, and we ourselves, as beings, at least have an "intuition of being", albeit, Aquinas argues, a "weak" (ibid., p. 52) one: we can make our way up to something we call God *analogically* from the very fact of our being (in Being). Such a proposition, however, would only be a proposition and remains, in the last instance, apophatic; for Aquinas, we can have a concept of God "only" as "inconceivable" (ibid., p. 52) precisely. There is no way that we might proceed from a concept of the essence of Being/beings to the existence of such a being. Caputo is quick to point out Tillich's misinterpretation of Aquinas, stating contra Tillich that "God is not a *particular* subsistent being (*ens*) but subsistent being (*esse*)" (ibid., p. 51). At best, we can make sense of God as an "infinite matrix in which we live and move and have our *being*" (ibid., p. 47); the act of being is an "uncontainable excess" (ibid., p. 47) in which all particular beings have their place and in which all takes place.

Yet it all starts with an "intuition of being", a term Caputo takes from Jacques Maritain, and which will—although discussed very briefly—be of continuing importance throughout Caputo's book. We are able to intuit "a nonessential perfection of *esse*" from out of the fact of our very being as, in some way, "a triumph of being over nonbeing" (ibid., p. 47), yet it "can be occasioned by anything from a single blade of grass to the implacable presence of the universe at large" (ibid., p. 47). All concepts and essences come too late to capture this sensing of existence. Esse then—our own received esse is no exception—is not the sheer fact of existence, not simply our "thrownness" in the world as Heidegger would say, but the result of our wondering at being, it is "the correlate [ . . . ] of an intellectual affirmation that something is, exists, rather than not" (ibid., p. 48), the recognition that all we see, feel and think has in some sense emerged out of nothingness and, for the time being, stands its ground. It is the realization that everything, in a sense, is full of being. Such proximity to wonder is what philosophy seeks: the recognition that our thrownness is no mere "that it is" but the extra-conceptual realization that being is, and has been *granted*, our being and that of all other beings.[1]

One could easily imagine someone coming up with the idea of weakening our weak intellects just a bit more: that someone is Eckhart. Eckhart, for Caputo, confronts Aquinas'

"edifying" apophaticism with a "more unprotected, bedeviling and anguished apophaticism" (ibid. p. 56): what if a being is not so much an upsurge out of nothingness, but a nothingness itself, dependent entirely, for its being, on God? Such a being would have nothing to boast upon, no intellect no matter how weak, no being that could be affirmed or granted apart from God, no speech that no matter how analogously could say anything sensible about God. If through the theory of participation some could already suspect some sort of panentheism in Aquinas (cf. ibid., p. 49), then such panentheism seems completed in Eckhart: the world is nothing if the world is not in God (cf. ibid., pp. 57–58).

Here, one will find a first radicalization: if Eckhart states that, if the "world would vanish [God] would be left standing", Caputo's radical theology sees them as "interdependent" (ibid., p. 58). Without the world, there would be no God—God puts Godself in play in the being of the world. Luther will radicalize Eckhart in his own time; if the Godhead of which Eckhart spoke was called upon to protect the divinity from the all too human speech that could cling to both reason and the claim to revelation, then Luther's *deus absconditus* initiated an anonymous rumbling—Caputo in this book quite often, and at strategic points, mentions Levinas' *il y a*—into the very thinking of divinity: it is by no means certain whether God is good. Caputo finds this "more haunting" (ibid., p. 65)—but we will need to be mindful and wary of Caputo's competitions looking for spookier, eerier accounts later on—version of Eckhart quite appealing. If theology comes up with a Godhead which would allow an anonymity to the point of a possible confusion between God and the devil, then this signals for Caputo, as mentioned, "an autodeconstructive disturbance *within* theology" (ibid., p. 66). It cannot name what it nonetheless wants to name.

Here, we find one more radicalization in Caputo, and one that is not that distinctively present, to my knowledge, in his earlier work: "might it be that the element in which we live and move and have our being is not concerned with *us* at all and this because it is not in the least bit *personal*? [Might] it be that what there is, is *indifferent* to us?" (ibid., p. 70). To be sure, there were long ruminations on Lyotard's notion of the inhuman and "the smile on the surface of matter in a remote corner of the universe" in *Hoping against Hope* (e.g., Caputo 2015a, p. 40) and on the fate of the human amidst cosmic entropy (Caputo 2019, pp. 216–26) and hints at such indifference too (Caputo 2019, p. 167) too. In his recent work, this indifference is best imagined, it seems, through cosmic dissipation, yet this idea of indifference, itself, seems to lead, as we will see, to theological consequences that Caputo does not entirely foresee.

If one could see some steps away already from God as a highest being in the metaphysics of Aquinas and Eckhart, such "post-theistic" thinking emerges fully only in modernity with Hegel and Schelling. Here, one might say, instead of being an entity supervising and steering the world from above, God *dissipates* entirely in the world. The first requirement of a radical theology, so to say, is met: God becomes another name for the world, and being's becoming and capacity for change is to be seen as the unfolding of divinity in the world through which God becomes God. Caputo applauds Hegel's post-theism and his supernatural reduction where "God died the day Jesus was born" (Caputo 2022b, p. 78) but complains about the radical "anti-apophatic result" (ibid., p. 93) of Hegel's concept of absolute knowledge describing the march of the spirit all the way to its realized eschatology in Prussia—just where Hegel happened to live!

This is why Caputo's earlier books already spoke of a "decapitated" or "headless" (Caputo 2013, p. 92) Hegelianism: this would be the Hegel who realized that all particular religions are but narrations or *Vorstellungen* of our encounter with the world, and its theological stakes (ibid., pp. 74–75; also Caputo 2013, pp. 88–92), but without the certainty that God's story ends well (Caputo 2022b, p. 97). A radical theology must here depart from Hegel. A new theme here emerges, whereas for Hegel thinking and being coincide, radical theology (although here already aligned with Aquinas' "paradoxical task" (ibid., p. 47) of naming being from out of a deficient intuition of being) gives up this pretense. "Hegel assumes that thinking being means being is thinkable all the way down" (ibid., p. 99), leaving no room for the unthought or for what Schelling will call the "unprethinkable".



The fact that not all of being is rational and thinkable allows to abandon the obscenities to which a Hegelianism is always headed: to state, for instance, that Auschwitz, too, "belongs to a larger divine narrative" (ibid., p. 99).

Radical theology cannot escape the problem of evil. Evil will no longer be thought as the lesser good within a greater divine narrative. It is for this reason, we believe that Caputo for the first time ever turns to Schelling who, in his earlier work, was rarely quoted and only made it into some passing references. Schelling, more so than Hegel, will allow a certain "groundlessness [ . . . ] inscribed in being itself, including in the being of God" (ibid., p. 103), making for the fact—contra Hegel—that there is a cleft between being and thinking. Caputo will see in Schelling an important "antecedent of radical theology" (ibid., p. 104), most notably in his idea of a "philosophical religion", which panentheistic traits takes seriously "the apophatic in religion" through "a deep encounter with the sheer fact of being rather than nothing" (ibid., p. 105). Caputo makes little of this parallel to Aquinas here, but it should not go unnoticed. Schelling's philosophy centers upon the idea of being as strife, will, and desire; God so wills the world because God lacks a world. In order to create world, however, God must overcome the ground and groundless of being and so call into being something other than Godself. There is a play of competing forces throughout Schelling's philosophy, not unlike the dyad promise-threat that we will note in Derrida's deconstruction.

Caputo will play out the "radical" Schelling against the "edifying" one, notably the one of the later *Berlin lectures* where Schelling, for Caputo, did no longer took any risks when reconfiguring the God of theism. What Caputo will learn from Schelling is a sort of "panvitalism" (ibid., p. 110) where the whole of being—creatures, creation, and God—displays a *longing* for being and existence, and where everything, God included, would need to be thought of as an "auto-engendering, self-birthing process of eternal becoming" (ibid., p. 107) where God, too, is on the way of becoming God. Such a "matrix" (ibid., p. 107) Caputo will also recognize in Tillich's ground of being. A second insight Caputo takes from Schelling is his view of creation as *pleromatic* rather than as *kenotic*: God is not first an inflated superbeing that only afterwards empties itself in creation, God's being is not elsewhere than in God's creation and only so finds fulfilment (through time and through finitude). This is also, perhaps, why we saw Caputo above willing to give up on the personal nature of the divine being.

There is, however, a separation between creation and creator in Schelling that will also haunt Caputo's radical theology. If God's yes to the ground, the calling into existence of its sheer groundlessness, is such that God can be seen as "the eternal success of existence overcoming the ground", creatures will have to deal more radically with both the principles of evil and of good, as, what is raised from the ground, say created, will always be, in Schelling's terms, "deficient" (Caputo 2022b, p. 116). Caputo makes a lot of Schelling's relapse into the safe haven of salvation story, but does not mention the exception that he here grants to the conception of the divine. It is, therefore, all the more curious that it just this exception that one will see recurring in Caputo's thinking of the call, of the insistence of the event. For, in Schelling, it is just this exception for the being of God that will make Caputo conclude, with Habermas, which he quotes often here, that God's immersing Godself in creation is but a "limited risk": the divine being itself is not at risk in its creation—"if the world itself were to perish, God would be still standing (and still longing)" (ibid., p. 115), just like we have seen in Eckhart.

Yet Caputo wants to push further: if God so loves the world, that God would be completely immersed in the being-of-the-world, would then not "a radical theology of the pure gift" be able to think a "greater love" (ibid., p. 115) of God? Caputo is enamored, too, by Schelling's overcoming of evil as privation of the good. Good and evil are opposing forces with which humans forever have to deal. Evil exhibits "the power of the unruly ground" (ibid., p. 127), a power that cannot be that easily domesticated by a metaphysics of participation. Here too, however, German idealism and its metaphysics needs to be "dialed down to a phenomenology" (ibid., p. 180), notably "to the stuff of a phenomenology of

ambiguity" (ibid., p. 131). This phenomenology, for Caputo, should lead once again to a serious engagement with God or being's indifference to the difference human beings think they make: what if it is strife, pain, and war all the way down (ibid., p. 136)? We must look, Caputo says, for a theopoetic rendering of the phenomenological rather than metaphysical credentials of such "sadness clinging to all life" (ibid., p. 120, quoting Schelling), for what if it all seems strife, pain, and war *right now*, but there still is some rest, joy, and peace—in short: happiness—*to come*? German idealism needs to be broken open onto a phenomenology of the absolute future (ibid., p. 137).

For now, we need to realize that this panvitalism, in which all aims for being and shares in a similar facticity of being, comes close to what we now call an "interconnectedness" (ibid., p. 144). Such interconnectedness, contra pretty much everyone in the philosophical and theological tradition, is thoroughly open-ended: our share of being and facticity make for an "open system"—a non-programmable programmability, Caputo elsewhere says (Caputo 2018b, p. 167)—which for Caputo is the very "source of the unruliness and risk in radical theology" (ibid., p. 141). The *very fact that we are* is for Schelling the "unprethinkable"—*das unvordenkliche*[2]—meaning that something cannot be thought ahead of his happening (what reason and thinking—in all of its modes: Greek, Latin, medieval, or modern—nonetheless often tried). The task of an "ecstatic reason", a beautiful phrase popping up regularly in *Specters of God* but already coined in *In Search of Radical Theology* (Caputo 2020, p. 28) is, once again, "paradoxical": "to bring the unprethinkable to thought [is] to catch the prephilosophical in the act without turning it into more philosophy" (Caputo 2022b, p. 141).

Thrown into such being, nothing is decided in advance and even God is left to God's unfolding (or not) in history. It is only afterwards, after the fact, that we might say, that, yes, this was truly divine. It might have been God. This is, for Caputo, Schelling's advance and one which pops up in plenty of Caputo's writings: "The existence of God will be determined by the test of unfolding historical experience. [Schelling] wants to know not whether God exists but whether existence proves itself worthy of God" (ibid., p. 149). Here then the question arises: "is God necessarily the necessary being?" (ibid., p. 149, also p. 197).

It is on this level that Caputo finds in Schelling's philosophical religion a predecessor of his own "religion without religion" (ibid., p. 153). Both now need, it seems, to be dialed down to a phenomenology and "feel about" for "the spiritual lines of force" that "penetrate" (ibid., p. 154) experience. The force of phenomenology here, one might suggest, is that this primal yes, this affirmation in and of being, this faith in what still might be to come, may not be able to be demonstrated (in and through a transcendental reason) but perhaps, more modestly, can be shown through an ecstatic phenomenological reason. For it is here, Caputo argues, when we "feel about" in our experience and are on the way on our "spirited search" that an originary "hermeneutic decision" (ibid., p. 157) needs to be made, namely the one that "[*takes*] history [ . . . ] *as* God's unfolding being" (ibid., p. 157). In these "depths" (ibid., p. 157)—a term that Caputo later in the book will, however, reject—no proofs are possible. It is the decision to *read* the world *as* the place in which God might happen, as full of sense at least, as filled with wonder and in need of our response, of our 'second' yes—a yes that seconds the first one, Derrida would have said, a little bit too late.

This decision, in Caputo too, remains enigmatic. It resembles Jean-Luc Marion who after pages and pages on the rationality of the gift and givenness in *Being Given* suddenly submits it all to a *vouloir bien voir*, a will and a decision to see "being" as "given" indeed (Marion 2002, p. 307). It resembles Heidegger's *Augenblick* in which Dasein momentarily faces its "existential solipsism" (Heidegger 1962, p. 233) confronting its finitude in anxiety. Caputo attempts to free himself from Schelling's decisionism in which the choice for good of for evil is a choice once and for all, so to say, and rid himself of this metaphysical baggage and comes close to Levinas' take on the Messiah, in which the "messianism" pertains to the subject alone, able to save (or not) all the others and the world.[3] The majority of continental philosophy here—and Caputo is unfortunately no exception—comes up with a thoroughly



*solipsistic moment*, even when for Caputo the "choice represents a kind of existential test of what we are made of, of whether we will *make ourselves worthy of the will within us*" (Caputo 2022b, p. 158). It boils down to the courage to be "life-affirming" or rather, "nonecstatic" and thus "life-denying" (ibid., pp. 159 and 158). We are free to make this choice, Caputo says, because such freedom is part of Being—it has a "*fundamentum in re*" (ibid., p. 160)—it is true that *there is* love, courage, and affirmation here and there—yet these do not oblige us in any way to choose for them and do not take away anything of our freedom; God, too, will have to trust the process. Yet the decision has a spectral quality about it—perhaps more than Caputo, in these rather obscure pages, allows: if we want to affirm that love is real, then we will affirm also that love is part of being. Nothing, however, can be proven here, so the reverse is "true" as well: if we want to affirm that blind destiny is real, we will affirm that such destiny is "part of being" too. No ultimate reason, no metaphysics, comes to our aid on this level.

What is clear, at least, is that there is an existential dimension to Caputo's "religion in the apophatic sense" (ibid., p. 161) and Schelling's early version of this: both represent "a certain rendering of the human condition that is available to everyone" (ibid., p. 166), it "is the issue of our unbreakable bonds with being" (ibid., p. 161). Religion broods over the "impenetrable mystery of the fact that there is being at all" (ibid., p. 168). We need to note that with this religion as "deep hermeneutics of facticity" (ibid., p. 177) Caputo has almost come full circle.[4]

For now, we need to see that, in and through the debate between Schelling and Hegel, Caputo finds a way to understand "the God of panentheism otherwise", a way "not opposed to the mortality of God, but in fact constituted by it" (ibid., p. 100). If there was too much pretense toward absolute knowledge in Hegel, Caputo finds that there is too little of a supernatural and ontological reduction in Schelling, as if God and being were an "agent cause, somebody doing something" (ibid., p. 174), whose activity, in Schelling's case, would be to will and love the world.

### 2.4. Radicalizing Postmodern Philosophy and Theology

For a theism transcended and a completed ontological reduction, Caputo turns to Paul Tillich. Caputo had discussed Tillich in great detail already in his *The Folly of God*, and there were clear anticipations of what we will call here the existential aspects of Caputo's work in a long note dedicated to Tillich's account of salvation in *The Weakness of God* (Caputo 2006, p. 332, n. 12).

Right at the outset, Caputo repeats that, with Tillich' s "theological critique of theology", we are at the very beginnings of theology (Caputo 2022b, pp. 187, 188). Tillich, not unlike Eckhart, Luther, and even Aquinas[5], looks for a "God beyond God" (ibid., p. 189), for which one perhaps finds in this book one of Caputo's best formulations of God to date—a postmodern reformulation of Aquinas' ipsum esse per se subsistens perhaps. The "God beyond God" is "inaccessible [because] it is the encompassing and hence unencompassable depths of being-itself" (ibid., p. 187, compare p. 50).

We have intimated already that, in this version of a phenomenological reduction, some things will have to go. For this, it is to be noted that one phrase of Tillich seems to have given Caputo more to think than others, namely that the idea of personal, superbeing overseeing beings of theism is "half-blasphemous and mythological" (ibid., p. 188)—the phrase pops up in almost all of Caputo's theological work. For Tillich, God needs to be reduced ontologically, that is, to say, to being. If God is to be compared to (but not identified with) the "ground of being" or as "being itself", namely as "that from which and into which things come and go" (Caputo 2018b, p. 291), we need to let go of all ideas that this divinity does things (or does not do them). God does not *intervene*, miraculously, or *reveal* God's being, in a clear and distinct message. Rather, if God is the ground of being, "the emergence of a world from God is not God's *doing* but God's *being*, not something which God *does*, which is piety's symbolic way of imagining it, but something which God is, which is the properly ontological way to think" (Caputo 2022b, p. 195).

Such a God, Caputo now insists, can only be known indirectly—through "particular beings, traditions, and cultural symbols" (ibid., p. 197). We can deny that the name of God is the best symbol for being-itself, but we cannot deny "being itself" as that of which this name is a symbol. If such symbols are, to some extent, interchangeable, "the real question is whether a given symbol of the unconditional is *worthy* of unconditional affirmation or whether it is an idol" (ibid., p. 197) and the real problem is to know the difference between these two, affirmation and idolization, since the danger always exists "to confuse [ . . . ] the mediator with the substance of what it is mediating" (ibid., p. 202). With Tillich, Caputo points to the cross, the cross of Jesus, as "a symbol of the self-effacing of the symbol in favor of what is being symbolized" (ibid., p. 197). The cross deflects from itself to point beyond itself. It is "a symbol of a symbol" and in a sense autodeconstructs itself: "it crucifies anything conditional that purports to be unconditional" (ibid., p. 205). Just as Jesus deflected from (the Kingdom of) God by pointing to the most ordinary things of life in which the Kingdom could happen (cf. ibid., p. 20), so too the cross tells us to not look at it directly but to what the cross, in a sense, achieved: to turn the lowliest into a symbol of the Most High. One can hardly hold against a hermeneut like Caputo to speak from and out of his own tradition—of which Caputo most certainly is aware, but still, it is difficult to imagine what exactly a radical *theo*logy would be without Jesus (if there is such a thing).

Tillich agrees with Schelling that an ecstatic reason is always and already "breached by the primacy of being over thought" (ibid., p. 200). As soon as we are, we are in being. Whenever we are in being, we are in God. Tillich's panentheism is such that "our being may be alienated from God, but God is not an alienated being. We are always accepted by being, even if we are not acceptable" (ibid., p. 198). The human being, for Tillich as for Caputo, needs to co-respond to this primacy and facticity of being by a "courage to be", and so echo the first yes of being (to speak like Derrida again). That I am rather than not, is a victory over "the forces of nonbeing" (ibid., p. 198) and this victory, in a sense, needs to be seconded time and time again, by the affirmation of hope and—why not—by a fidelity to being. If we do not want our lives to be "flat" (ibid., p. 200), whether this be through calculative reason that makes all things alike and uniform or through the utilitarian bourgeois that Tillich critiqued because of its hopeless pragmatism, we need to be in tune with this "mystical element" (ibid., p. 200 full circle now: Caputo 1978), with "the sheer mystery of the that-it-is-rather-than-not" (ibid., p. 201). This element can be found in art and science just as much as in religion today perhaps more in the former than in the latter. It just needs the "phenomenological realization of the absolute within and all around us" (ibid., p. 200).

At this point, the lines between Tillich and Caputo have become blurred. It is easy to see that, for a long time, Caputo has been enamored with this kind of panentheism. This, perhaps, is why it takes him a long time, too, to take his distance from this kind of post-theism and perhaps even the vitalism that accompanied it through Schelling. We now need to see Caputo turning away from Schelling and Tillich. This distance was announced already in *The Folly of God* (Caputo 2015b, pp. 67–68) where Caputo realized that the non-human back-up of Schelling (and its *height*) in some sense survived in Tillich's account of the *depth* of being as a stabilizing ground.

In the following chapters, Caputo sets out the contours of both the idea that radical theological is inherently a political theology and of his farewell to the language of ontology (which is a farewell to Tillich). To speak of the unconditional, like both Derrida and Tillich do, in effect "is not a matter of strictly theoretical interest. Nothing matters more than matters of ultimate and unconditional concern in the conduct of conditional [and] practical affairs" (Caputo 2022b, pp. 213–14). Here, the difficulty of a phenomenology of ambiguity shows itself: not only every act is immersed "in a sea of competing potentialities" (ibid., pp. 219, and 272), in promises and threats, its symbols too have to deal with the "ambiguity of a mediating term: the unconditional is never present except under concrete conditions [with] which is prone to be confused" (ibid., p. 214). This already is enough to realize that the "ground" is utterly "groundless" (ibid., p. 229) and that there, thus, is an "excess in



every given order" (ibid., p. 232), a non-programmable digit in every program. Whatever is grounded, will be so only for the time being, and will sooner or later be shaken in its very foundations.

To keep the system open, to always have faith that the truth of the system may not lie within the system, these discourses, say the one of democracy, rely on a memory and a promise: it is to both mourn that democracy never really was it was meant to be, or what it could be, and that, in our current democracy, there still lingers this promise to make the best democracy happen (cf., ibid. p. 251). It is this "between" of a mourning memory and a hopeful promise that will make it possible to "inhabit the distance" between the current order and the order to come. Caputo is quick here to tone down Tillich's ontological language: "what is called 'the ground of being' is ultimately the structure of temporality" (ibid., p. 251). It is *here and now* that the call of the event will have to be answered and we will mourn the passing of this here and now as soon as we start to act—"I could have ... "; "I should have ... "—as much as we realize that our yes already echoes a first "abiding affirmation", "a wider embrace of the world" (ibid., p. 239) than any ethical or political stance we take—otherwise, why act at all?

With such a "hauntological reduction", we are of course already on Derridean terrain. It is instructive to see just how Caputo sees such a Derridean radicalization of Tillich, for the latter's "panentheism conflates the unconditional with the ground of being, [underwriting] the unconditional with an ontological insurance policy which avoids the risk of a radical groundlessness" (ibid., p. 243). The ground of being then should give way to an "unprogrammable event" as the depth of being does to a differential play and its disseminating effects (ibid., p. 244).

It is here, more than anywhere before, that we see Caputo take leave of ontology: "if I resist the language of being, it is not because I favor nonbeing, and it is not because I do not think that being is good. [My] reservation is that ['being'] has a way of settling in place and meaning immutable presence, and God is framed as some sort of Deus erectus, stiff, unbending [and] unchanging" (ibid., p. 247). We can understand why Caputo's theology becomes a political one then, since if there is one discourse where rigid identities and immutable pasts and presents pop up, it is, in effect, in politics. What we still need to understand, however, is what comes in being's place?

Caputo might be right that the language of being seeks too much alliance with the classical idea of an inner essence leading to an outer manifestation (ibid., p. 219 and p. 248)— as if any such essence would lead, straight away, to any such existence. There is more than one line between A and B. In its stead, as we have seen, comes a "sea" of conflicting possibilities in which things unfold "laterally" rather than "vertically" (ibid., p. 248): they do not come from a mysterious depth but are, in a sense, free-floating on the surface. There is not just one meaning to these things. The philosopher and theologian of the event just gives "progress report[s]" (ibid., p. 248) what has happened; he or she does not wrest any essence out of a mysterious material cloth. Things do not have an essence, Caputo further argues, "they have a history" (ibid., p. 248). The apophatic imagination, then, wants to be "loyal" (ibid., p. 254) to this history and, therefore, offers up "a phenomenology of historical life" (ibid., p. 254)—Heidegger again—in which no super being, no ground of being, and finally no God beyond God is presupposed. Caputo here, with this search for the "elemental power" (ibid., p. 238) stirring within religious, cultural, and scientific traditions, once again rejoins his earlier work on Heidegger with whom Caputo agreed that the overcoming of metaphysics "wanted to think that from which the entire tradition springs" (Caputo 1982, p. 231) by taking up the entire tradition as our historical tradition precisely.

The unconditional, then, turns out to be a ghost: it wants to be, but it is not yet, not entirely. "It *calls* for being", it is the idea of a "possibility" (ibid., p. 255). And this spectral call is all what remains of the spirit after the secularization of the holy spirit (Caputo 2018b, p. 300; 2020, p. 12). After the secularization of the spirit, however, this specter still blows wherever it wills. It calls for us, though, it relies on us "to make things better" (cf. Caputo 2022b, p. 256) because things *can* get better.

The call does not say, however, how we can make things better; it sends neither messages nor revelations. It is not strictly formal though, since it *presences* within our traditions. Caputo is clear, however, that this call remains transcendent to these traditions; once again, the truth of the system lies outside the system itself. It calls for human beings to be "ready for the coming of something very strange" (ibid., p. 259). Such prophetic imagination, Caputo tells us quoting Tillich, "is transcendent. Rational expectation is immanent" (ibid., p. 259) and although Derrida will not use these words exactly, Caputo tells us that Tillich's take on such transcendence is "strikingly similar" (ibid., p. 259) to Derrida's position on the absolute otherness of the future.

Here we are, again, at the very beginnings of theology, at the sources and elemental power inspiring "a more unconditional theology" (ibid., p. 263).

It is here, too, that this theology faces situations that force Caputo to speak almost like Blumenberg in modern times: new questions have arisen for which the old answers no longer suffice—Caputo even speaks of a "new axial period" and a "seismic shift" (ibid., pp. 283, 297). This situation, for Caputo, seems to be the "facelessness of the posthuman", in which we see Caputo waver for a first time. After applauding in his other work, the rise of quantum science, after sticking to a "lively" vitalism for a long time, and his long love for panentheism, the posthuman now obliges him to somewhat abandon this panentheism, its vitalism, and the enchantment quantum physics brought along for many thinkers of today.

Caputo reminds us of the "cold hermeneutics" (ibid., p. 267) of deconstruction, in which no meaning takes precedence over any other, and the autodeconstructive character of *différance* turns the personal and interpersonal into but "effects of impersonal forces" (ibid., p. 267). Against the Gadamers and Ricoeurs of this world, Caputo acknowledges that these meanings cannot be appropriated, but drift and disseminate—things "come together" for a while, only to autodeconstruct after. The good news, if any—we still will have to see—is that that an autodeconstructive system is, just because of its "irreducible unruliness" (ibid., p. 268, notice Schelling's wording), open-ended. No one can lay hold of it or make any claims toward complete comprehension. Only a "non-programmable programmability" can resist the totalization of technology and its codes and so "preserve the play" of meanings (Caputo 2018b, pp. 167, 269). There is just this "sea" of "competing possibilities" (Caputo 2022b, p. 272), in which things come together for good *or* for bad and which is "a contest" the thinker should describe, witness without doing "any injury" (ibid., p. 272) to. Whereas one might have expected, by now, Caputo to come up with a "poetics of the surface", he instead seeks to "displace a metaphysics of the deep with a poetics of the deep" (ibid., p. 273).

Caputo is not one to shy away from the challenges of the future, and in its final part Caputo's book at last comes to the "*eeriest* form"—notice the superlative and wonder about this competition for the spookiest of guests again—of the apophatic imagination (ibid., p. 277). Caputo had written before on the challenge of AI, of cyborgs, of quantum physics, but now takes on two last hurdles: the phenomena of "the upload" and the Big Crunch—the fact that our sun, it seems, is finite too.

Whereas Donna Haraway's cyborg "keeps one foot in the body" (ibid., p. 282), welcoming the mixture of human and non-human non-nostalgically, Caputo hesitates when it comes to the phenomenon of the upload. This hesitation is revealing, however: for one thing, it reveals Caputo's confession to not-knowing where it all leads to the very end. The upload abandons all things human by "uploading" the brain into the cloud and so rid ourselves of all embodiment and decay. With this, Caputo argues, we have entered into "the most extreme case imaginable of the uncanny", something very *unheimlich*, alien even (ibid., p. 290). "Spookier still" (ibid., p. 292) is the thought that perhaps we are already living in a simulation and in a virtual world. The theological resonances of the upload are, however, not lost on Caputo: the thought of overcoming death is, in effect, not foreign to the theological tradition and Caputo's comments on the resemblance between medieval angelology and postmodern simulations are intriguing. Yet he also notes that theology, up until today, has always sided with the body and questions whether there would be

any theology left if it is not in any way "tethered to biology" (ibid., p. 298). Caputo in effect ends this discussion by admitting that the upload is "far from being clearly desirable" (ibid., p. 304). One should not fail to notice, however, that, because of the up bidding of the spookiest and eeriest form of the call and its contemporary figures, Caputo here leaves us in an aporetic situation where the "beginning" and the "end" of theology more or less coincide, leaving us to wonder whether there are "good" spooks and "bad" spooks.

Still, the "spookiest specter of all" (ibid., p. 304) is yet to come, and one can, in effect, "hardly imagine an eerier thought" (ibid., p. 305) than the one which states that our cosmos is destined to disappear. Caputo makes it clear that the Anthropocene, and the concomitant climate change, is a problem indeed, but the universe itself facing extinction and total entropy is a sheer mystery (cf. ibid., p. 306). This spells the end of panentheism which tied the fortune of God to being and the cosmos precisely. "This entropic destiny radicalizes the reduction" (ibid., p. 390) once more; we will have to deal with the existential questions of finitude, not only of our finite being, but also of the universe as a whole.

### 3. Transcendence and Happiness: The Existential Caputo

#### 3.1. Finitude and Mortality

Caputo will insist that there is joy here: "we affirm that all there is, is a staggering and glorious mystery, the glory being to have been offered a part in its cosmic play" (ibid., p. 310). Caputo speaks of a sort of decentering here, a "cosmopoetic correction of the phenomenon of reflexivity" (ibid., p. 310). Whereas, for the larger part of the philosophical tradition, thinking thinks itself—where what we think *reflects* ourselves back to us—in the facing of this mystery of our place in the cosmos we think because something has already given us to think "before reflection arrives on the scene" (ibid., p. 310). If we inhabit the distance between being and thinking, it simultaneously is true that "we humans are one bit [ . . . ] of the world where the world has begun to worry about itself (ibid., p. 310). Again, such a hermeneutics of factical life "keeps making the mystery that such a world is all the more mysterious" (ibid., p. 312).

Such a mysteriousness of being part of a finite universe, of being one gaze in which the universe gazes at itself, comes with its own existential virtues. It requires "sensitivity, insight and discernment" (ibid., p. 314) to learn to appreciate anew "the religious dimension of our lives" (ibid., p. 314) amidst the frailty of our being-in-the-world.

What *chances* are there for a God tied to a finite cosmos, after the end of panentheism? Caputo advances the thought of a "mortal God", one who, as Derrida somewhat prophesized, "deconstructs himself in his own ipseity" (ibid., p. 318) without, that is, sovereign power. The fortunes of such a God are bound entirely to the response humans give to its spectral call. The name of God, in this story, "is not the name of an entity, but an experience, not a thing but a relationship, not merely a thought but a deed" (ibid., p. 316). In the absence of God, we are to do God's work. "It falls on us [ . . . ] to make ourselves worthy of what is of unconditional worth" (ibid., p. 320).

For this, we need to see our world as Eckhart saw the rose. To see the world without why is to reconfigure the apophatic imagination (cf. ibid., p. 322) and to clothe the world with a "zone of unconditional respect" (ibid., p. 323). Heideggerian worlding of the world thus requires a "living without why" (ibid., p. 324): a confession of unknowing and deep respect for its coming and going, a prayer about its whence and whither. With this prayer to the promise of the world, we can, Caputo finally states, "put to rest the objection that 'radical theology' is a strict academic enterprise of no interest to people who are interested in 'real' religion" (ibid., p. 324) and, yes, it can also be preached—one needs to turn our contemporaries away from their "flat", calculative lives.

One might, in effect, argue that we must *learn* to see the world without why. Whereas we are *used*, if not addicted, to treating things as resources to be optimized, it perhaps takes a "theopoetic attitude" indeed to see that "what makes something precious is precisely its perishability" (ibid., p. 328). Theopoetic prayer and preaching "affirms" the world, and

everything in it, "for the while that it is" (ibid., p. 330) and will so try to find an "austere joy" (ibid., p. 14) in its mortal God. One must, perhaps, imagine the radical theologian happy.

### 3.2. The Affirmation of Life and the Smiles to Come

"Precious" is the world Caputo will use in all of his books to denote our mortal condition (e.g., Caputo 2013, p. 227; 2015a, p. 176; 2019, p. 63; 2020, p. 93). It is on this word that we will dwell in this second part of our essay, for Caputo's "nihilism of grace" (Caputo 2022b, p. 328), however beautiful, raises some important questions. These questions revolve around the idea of community over and against Caputo's existential solipsism noted earlier and, secondly, around what we could call a postmodern version of deism over and against Caputo's rethinking of panentheism. Caputo's version of such nihilism was already elaborated in *The Insistence of God* and in *Hoping against Hope*, although it is here, well, deepened, by the idea of a finite cosmos.

The joy in the mortal God of which Caputo speaks is the joy we receive from doing God's work, considering that God depends us on us to be called into existence. If no superbeing is up there in the sky, then all we can do here is to act as if God ruled the work, to follow the folly of the Kingdom as best as we can. It is from these works that a certain joy stems, albeit a joy that befalls us, not despite, but in and through and along with all the suffering (e.g., Caputo 2022b, p. 328). Our mortality, Caputo tells us, "intensifie[s]" (ibid., p. 331; Caputo 2015a, p. 159) our lives. In and through it, we penetrate the experience of "the call of temporality" (Caputo 2022b, p. 251) a bit more. Our lives tend to be a little less flat when we realize that the "unconditional [is] time"—nothing more, nothing less (Caputo 2015a, p. 33); it is because of time that there is urgency. It is from this too that hope springs—a hope without non-human back-up—and that there is "ignited the great compassion of mortals for their fellow mortals", a "community of mortals", as it were, which is all too easily forgotten in our everyday calculations and practices (Caputo 2015a, p. 161).

Yet Caputo speaks little of community. To be sure, the joy in the mortal God comes with acts of compassion, with "works of mercy". Caputo is quick to point out that, in the absence of God, it is up to us to do God's work. Even more, the Kingdom of God is nowhere else to be found than in and as these works (Caputo 2015a, p. 62). We do these works not for the sake of a heavenly reward—which does not exist, or even insist—but entirely for the sake of the other. Facing the pain of the other, we encounter something that merits our attention unconditionally. Our assistance to the other is the response we give to the call: we do what we think we need to do to make the world look were God to rule the world (Caputo 2022b, p. 20). Caputo's stance is more and more an axiological one: how I can make myself worthy of the call? How am I worthy enough to assert myself in this world at all, considering that, absent any and all god, everything is permitted and I am free to decide "the sort of being [I] want to be" (ibid., p. 198)—except that "being", or the call, or *God knows what* already has made an offer that we cannot refuse.

Commentators have made little of Caputo's repetition of Derrida's take on affirmation.[6] Yet there is a lot at stake here. Just consider Caputo's casual commentary: "The unconditional always already lays claim to us before we open our mouth. The way Derrida puts this point is to say that whenever I *decide* that is the *decision of the other in me* [ . . . ] When I say yes, that is the second yes, the first one having always already embraced me" (ibid., p. 232), and grants me a stance in the "wider embrace" of the world. We are *spoken for*, a certain Heideggerian phrasing would say, except that here we are made available by precisely not being available to ourselves (entirely). The hauntological call breaches our grasping concepts to such an extent that we are always already *given over* to our being in the world, to others and to otherness.

This encourages Caputo to state that the "right response to life is affirmation" (Caputo 2019, p. 263): this 'yes', is "a way to value, prize and treasure something" (ibid., p. 263) our being, for instance, and that of others. It is here that the base-line of Caputo's existentialism can be found, both with regard to the self and with regard to the other. For it is this

affirmation that sparks hope, and a certain happiness perhaps; this hope is not the hope of an afterlife, but the simple hope for "more life", not because the future will be better, but because there always is future and time. This future is not per se better (as Caputo in his more cheerful moments is inclined to write) but rather is "always worth more" (Caputo 2020, p. 92).

Here, then, is one thing that one can learn from Caputo's theology. If Albert Camus was right to say that the only real problem in philosophy is the problem of suicide, then one might imagine Caputo's answer as follows. Radical theology circumscribes the wonder of the fact *that* we are and that we "live on" in face of adversity, *no matter what*. We "live on", too, not only for "big" ideas such as the democracies and justice "to come" but also for those little pleasures, the friendships and the loves *still* "to come"—who, after all, know what life still has in store for us? To the one giving up, then, Caputo's advice would be to *give oneself over* one more time even if there is just one *smile* "to come".

Radical theology only wants to convey that this life, and its search, is not without religious significance. One might wonder, though, why the traditional answers of confessional theology would not suffice—from whence indeed the search for the "sources" of religion? Because such confessional religion, if it is honest, would admit that these answers do not always suffice. This is death's intensification of life: *words fail* in our "companionship offered to the dying" (Caputo 2020, p. 75) and we, in more than one way, fall out of our confessional and traditional narratives—it is not always appropriate to console the dying right away with a thought of an afterlife for instance—and the distance between a radical and confessional theology is opened. If Derrida said that he was "never more haunted by the necessity of dying than in moments of happiness and joy" (Derrida 2007, p. 110), one might imagine Caputo saying that one never is more alone than when tending to the other. It is indeed hard to see how the confession of not-knowing (what will come, what to say) can genuinely reach the other—it is as if the existential solipsism haunts Caputo even when he is closest to the other, especially when considering, like we did above, the total absence of reasons in which the hermeneutical decision concerning the call is to be made. Here, too, no word from another would come to our aid—for Caputo, on this score, a simple *"It is fine. Just be with me"* will not do here.

### 3.3. Community and Solipsism

There is thus a legitimate concern that Caputo somewhat overburdens the human being in his account of these works of mercy—a suspicion stemming from his reliance on Levinas perhaps. There is a sense in which we all remain *lone interpretative islands*, and like in 1997, all just blind men and women with a stick (cf. Caputo 1997, pp. 313–14), just seeking our way. This relative absence of the question of community in Caputo is noteworthy. If we mentioned above that Caputo dialogues with "everyone who is anyone" in contemporary philosophy, one cannot *not* notice that Caputo never really interacted with Jean-Luc Nancy whose thought of the other as a "companion" (Nancy 2014, p. 141) might be very helpful here—an absence all the more remarkable because Derrida, in a sense, never stopped dialoguing with Nancy.

A similar absence of genuine community can be found in the examples Caputo uses of our encounter with the suffering other. These examples are often examples of simple care for the other; one must not forget that Caputo, early on, already brought in the *heart* against all the Heideggerian bombast (Caputo 1993, pp. 61–74). Caputo will often point to Martha's example, taking care of Jesus' bodily needs. In his 2018 book, however, he paints a Derridean pictures of nurses in pediatric oncology he once gave a lecture to, nurses who respond to the call of each and any other brought in without there being a big master plan. Although they had to follow the rules of medicine—the force of law—each patient is in any case other and unique and demands that justice be performed to his or her singular case. Here too, therefore, justice suspends the force of law even if only for a bit. The face of these others is in each case singular and forces them to act in a singular way when all the knowledge they have is a necessary but never sufficient condition. *We do not know*

*what works, but let's hope for the best*. "They lead lives of *overexposure to the event*, of constant visitation by emergencies" (Caputo 2018b, p. 229). Though the suspension of the rule is here difficult to deny, one must, however, still question whether their institutions—their community—does not in any way protect against such constant overexposure, whether or not the Levinas-like hyperbole is here properly placed. These nurses, too, enjoy holiday leave, and have routines that curb the calls coming from everywhere. Instead of the force of law as a hindrance of justice, something that needs to be removed albeit temporarily, these communities and its practice, can also be seen as an opening onto such calls and exposure. These nurses are *embedded* within an institution and education that prepares them the best it can for such overexposure. There is not just the "next case" (Caputo 2018b, p. 225), the next case is always and already *one more* case. What Caputo describes as isolated acts of companionship are in fact very often embedded in a community and an institution. We have found, earlier, that Caputo dares to stretch the distance between the conditional and the unconditional (Schrijvers 2016). In this case, this would mean that the next, singular case is played out in favor of one more case, like the priest I once asked whether he ever got used to the Eucharist responded that celebrating the Eucharist, too, can become a habit.

Other examples of Caputo of the entanglement of life and death confirm this solipsistic, somewhat individualist, stance and limited account of community when speaking about the affirmation of life. To be fair, relatively late in *Specters of God* the idea of community does pop up as if Caputo realizes that a "more communal structure" is still missing from his account of joy (cf. Caputo 2022b, p. 331). We will mention two examples of this joyful but lonely affirmation of the event of world in Caputo. First, there is his secularization of the idea of resurrection. Secondly, we draw attention to an odd phenomenological intuition of a family photograph.

Resurrection, for Caputo, has become a reprieve, as when Cixous after Derrida's diagnosis craved for a bit more life (Caputo 2013, p. 231) and Lazarus' sister does not really care about an eventual afterlife but just wants "more time" with her brother (Caputo 2006, p. 236; 2020, p. 147). Another example shows that Caputo, on this score, never seems to leave the circle of friends and family at all—not attaining, once again, an idea of community. In an intriguing observation when speaking of the "unconditional natality" of the child, as a temporary victory of life over death, Caputo mentions that it is to celebrate this victory that "when two or three [ . . . ] generations are gathered together, we rush for our cameras to get a shot of this moment, made infinitely precious by its transiency" (Caputo 2019, p. 63). This may very well be the case, but a thought of community would need to show how such transiency extends beyond the borders of the family.

We noted above that Caputo's reduction is not exactly a phenomenological striptease: reducing religion to all its purity by purging it from supernatural entities to come up with the "naked" that it is of a "protoreligion" (Caputo 2022b, pp. 178, 238). The protoreligion is not the ideal, and bare, essence of material religion. There is no purity here, and the question that will keep haunting Caputo is *how much of a remainder* of confessional religion there is in this protoreligion. For the time being, the supernatural reduction seems to offer a way from the confessional tradition to its radicalized version but it is not sure whether there is a way back to the boundaries of confessional theology. This does not seem to be so much a problem for Caputo, yet it leaves the status of the impure "remainder" of confessional theology somewhat hanging. Our guess would be that this remainder is glaring with the problem of community. Radical theology has no other bonds with the other than the fact that we are all somewhat isolated islands figuring it out by ourselves, interpreting our searches for meaning all separately. Yet one could have expected that once we are trying to "inhabit the distance" between confessional traditions and radical theology there is no more room for any depreciatory remarks toward religious communities and the concomitant "long robes" (ibid., p. 239, p. 314, . . . ). These, too, should not be simply seen as a hindrance for religion's radicality, or as something that needs to be removed, albeit temporarily, these communities and its practices can also be seen as an opening onto such calls and exposure.

### 3.4. Good Spooks, Bad Spooks: Metaphysics and Panentheism

But cannot one not just disregard this call and pass by the other in need? One certainly can, and most of the times we do, but we cannot do this without guilt. It is here we find Caputo repeating some crucial features of the "ontotheological imaginary" (ibid., p. 40) and so his radical theology still seems to be an heir to a panentheistic metaphysics. When discussing the facticity of being, for example, Caputo mentions that this facticity happens like a Derridean gift, in which, because there is no identifiable giver, "the recipients are not put in a permanent debt" (ibid., p. 177). Yet, just a few lines further, Caputo observes: "if we are not religious in the sense of the protoreligion [ . . . ] we are condemned to lead trivial lives [and] fail to make ourselves worthy of what is happening to us" (ibid., p. 178). Having received the call, these trivialities, one might say, make us *sin against the event*: as sin was usually thought as the privatio iustatio, as the absence of salvation there were salvation was nonetheless offered, here sin seems to be conceived as a privatio eventus, the absence of that which ought to be there and, all things considered, is always already there—exposure to the event. This "exposure" is an effect an offer one cannot refuse.

If one starts to think along these lines, all of a sudden, a lot of ontotheological gestures surface in Caputo's work. There is, on the one hand, the *constant presence*, not to say *stetige Anwesenheit*, of the event and the call which is quite awkward considering that it ought to be "without sovereignty" (ibid., pp. 243 and 251). Why would something that is not sovereign be around all the time? If it calls always and everywhere, does it then not hover above us as a specter of a sovereign God—is this a "good" or "bad" spook?—would do? There are a few indications of this constant presence of what insists, when Caputo for instance mentions that only the conditional exists but "in continual unrest and exposure to what insists" (Caputo 2015a, p. 108) or when he describes Derrida's notion of the underlying call "by which *everything* we do is driven, in politics, ethics, [by which] *everything* is solicited, in response to which they rise up" (Caputo 2020, p. 89). For all the non-programmability within the program, this depiction leaves us with a call that hardly can be escaped and that seems *as totalitarian* as once the Stasi was. Suddenly, then, some things fall in place. So, for instance, the exceptional status Caputo tends to give to this call: whereas all and everything in this "sublunary" realm deals and has to deal with the entanglement of life/death, with the Derridean dyad of the promise and the threat, things are different in the "supralunary" realm of the call. This call, apparently, "calls *without promises, without threats*" (Caputo 2022b, p. 263, my italics). Whereas the created order was caused by an itself uncaused creator, here the order of the world undergoes the laws of the promise and the threat by a call itself calling immune to all threats and promises. Sovereign is he (or she or it) who can make an exception of himself.

We are being too critical. Caputo offers us a theology "worthy of the name", one that asks all the questions at the right time, by which Caputo truly shows himself a thinker of today. Yet this lingering presence of some ominous ontotheological spook might perhaps mean that this constant presence, too, must fall. What if the call is not present "always and everywhere", but *presences* here and there? In this case, we would no longer deal with a heir to panentheism, but we might end up with a postmodern version of deism, in which God is silent, hidden, cornered, so to say, in the beginning of his creation with a few miraculous *presencings* not to be excluded. Such deism would not deny the presence of the call; it denies the constant presence of such a call. Such deism would perhaps also deny the spooky and spectral presencing of the call ("appearing without presence", Caputo 2020, p. 51) in favor of a *presencing without appearance*, but would not deny that the transcendence at issue here, even when it cannot be contained by the world, can at least be *conserved* by the world, in its traditions and communities—even if only for a little while (Schrijvers 2017). This transcendence of love stretches out from the simple fact of (always) being with the other, holding him or her, all the way to holding on to the presencing of love to all beings *within* our world.

**Funding:** This research received no external funding.

**Institutional Review Board Statement:** Not applicable.

**Informed Consent Statement:** Not applicable.

**Data Availability Statement:** Not applicable.

**Conflicts of Interest:** The author declares no conflict of interest.

## Notes

1. Maritain's concept of an "intuition of being" is as such absent from Caputo's earlier work on Aquinas. Nonetheless, a similar line of reasoning can be found there, see (Caputo 1982, pp. 115–17, 268).

2. It is quite possible that this concept sparked Caputo's interest in Schelling as it is expressedly mentioned at (Caputo 2019, p. 234) as part and parcel of a "hermeneutical ontology" trying to make sense of the element in which we "move and have our being".

3. See Levinas (1983, p. 120), "Le Messie, c'est Moi. Etre moi, c'est être Messie".

4. See for instance (Caputo 1993, pp. 50–52), looking for a "hermeneutics of jewgreek facticity" in Heidegger.

5. For this, see (Caputo 2015b, pp. 15–16).

6. Derrida's take states that thinking in a sense *follows* this yes, just as Caputo here has argued that "thinking always comes after being". Whereas thinking (a second yes) tries to comprehend or even just understand the first yes—which is indeed very much like an offer one can't refuse, for good or bad—this first yes is already in advance of any response or answer; it *laughs* at every attempt to think it through yet "asks nothing but another yes". For Derrida, see especially (Derrida 2013, pp. 70, 77, quote is p. 74).

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
