# Peer review of "Keeping Up with Caputo: Of Specters and Spooks—Transcendence and Happiness in Caputo’s Radical Theology"

_religions, doi:10.3390/rel14040550_

Round 1

Reviewer 1 Report

Overall, the paper is quite good: it provides a useful introduction to Caputo via his most recent book, and therefore provides also a good introduction to that book. However, the article sometimes reads more like a book report than an article--there are long passages that seem to simply summarize the book, and I found this to be, at times, distracting from the overall arguments/theses of the article. It would be good to tighten the paper a bit (especially section I). This would leave more time to flesh out section 2, which currently is quite underdeveloped, but seems, to me, the more promising of the 2 sections, in terms of originality and adding something new to the literature. The critique of Caputo's individualism seems correct to me, and I'd like to see that fleshed out a bit more. 

Author Response

Dear, 

Thanks for the review. I agree that section I can be trimmed a bit and have done so accordingly. However, the essay is intended to provide an overview of Caputo's most recent work and connect it to this earlier work. In this regard, I think it might draw quite some readers to this essays - as it is the first to discuss Caputo's recent work extensively. 

I am happy that the author shares my critique of Caputo's individualism and absence of community. I offered, however, everything that at this point I can say about the matter but can perhaps extend my own critique of Caputo (as lacking in sheer 'presencings' of the call) a bit.  

Thanks again - it is always difficult to discuss a recent work like this and am quite happy with the thoughts of the reviewer. 

Reviewer 2 Report

This is extremely well done, a perceptive study, and the criticism at the end is interesting and helpful to me.

Author Response

Dear reviewer, 

Many thanks for this review: I am happy that you found my critique thoughtful. 

Reviewer 3 Report

This is an important article. It has few flaws, and brings the potential of inspiring much fruitful work at the cutting edge of the very disciplines of philosophy of religion and systematic theology.

There's not really much to contend with here. The critical engagement with Caputo's positions could be a bit tougher on him, and profitably include an explicit connection to the works of mystics in the Western tradition, such as St. John of the Cross or St. Hildegard of Bingen, whose positive apophaticism would function as a helpful corrective in criticizing Caputo. An engagement with the content of non-Western philosophical and religious traditions could also help contextualize the perhaps somewhat monolithic framing of a normalized post-modern condition, but these are not major flaws.

Line 870 & 959 lack titles in the references.

Author Response

Dear reviewer, 

Many thanks for your kind words. I agree that I, here and there, could have been a tougher on Caputo (especially with regard the metaphysical remainders in his postmodern panentheism). It is, however, clear from reading this book that it Caputo sees it as a sort of 'farewell' to his readers (to which my own remarks on 'full circle' in a sense alluded). In that regard, I didn't want to be too harsh for him - it is quite the accomplishment to write a book like this, especially considering he is retired. 

I agree that a comparison to mysticism and non-western traditions would be in order. That would be however the topic of another study. I think a thorough rereading of his earlier work on Eckhart would be helpful for this. But, as you say, there is much to explore here with regard to a more positive apophaticism. 

Thank you.